# Understanding the Effect of Dining and Motivational Factors on Out-Of-Home Consumer Food Waste

**Francesca Goodman-Smith [1], Romain Mirosa [2] and Miranda Mirosa [1,*]** 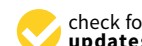

1   Department of Food Science, University of Otago, 362 Leith Street, North Dunedin, Dunedin 9016, New Zealand; f.goodmansmith@gmail.com
2   Quality Advancement Unit, University of Otago, 362 Leith Street, North Dunedin, Dunedin 9016, New Zealand; romain.mirosa@otago.ac.nz
*   Correspondence: miranda.mirosa@otago.ac.nz

**Abstract:** Approximately 12% of total food waste is generated at the hospitality and food service level. Previous research has focused on kitchen and storeroom operations; however, 34% of food waste in the sector is uneaten food on consumers' plates, known as "plate waste". The effect of situational dining factors and motivational factors on plate waste was analysed in a survey of 1001 New Zealand consumers. A statistically significantly greater proportion ($p < 0.05$) of participants reported plate waste if the meal was more expensive, longer in duration or at dinnertime. Irrespective of age or gender, saving money was the most important motivating factor, followed by saving hungry people, saving the planet and, lastly, preventing guilt. Successful food waste reduction campaigns will frame reduction as a cost-saving measure. As awareness of the environmental and social costs of food waste builds, multifactorial campaigns appealing to economic, environmental and social motivators will be most effective.

**Keywords:** food waste; food service; motivators; dining factors; cost savings; sustainability messaging; communication; interventions

## 1. Introduction

Food waste is an issue of great concern, with around one-third of food produced not eaten [1]. The issue is gaining traction, with the United Nations Sustainable Development Goal Target 12.3 (SDG 12.3) to halve food waste at retail and consumer levels by 2030 ratified by 193 countries [2]. Governments representing 50 percent of the world's population have set an explicit national target in line with SDG 12.3 [3]. In addition to the USD 1 trillion of economic costs food waste totals globally per annum, environmental costs reach approximately USD 700 billion per annum, and social costs amount to USD 900 billion, representing a real cost of USD 2.6 trillion annually [4]. Reducing food waste is a practical way to mitigate economic, environmental and social issues concerning the global food supply [5].

Increasingly, countries are reporting the quantity of food waste generated at different stages of the food supply chain. In 2019, governments representing 12 percent of the global population were measuring food loss and waste [3], and the percentage of countries measuring this will increase significantly following new obligations introduced in European Union (EU) waste legislation in May 2018 that require EU member states to monitor food waste levels at each stage of the food supply chain as of 2020 [6]. In developed countries, around 61% of calories are wasted at the consumption end of the food supply chain [7]. Consumer food waste includes food wasted in the home as well as food wasted out of the home [8]. In Italy, approximately 35% of food is estimated to be consumed out of the home [5]. While household food waste has been researched extensively, there is still limited literature

that focuses on out-of-home consumer food waste [9]. Due to an increasing proportion of food being consumed out of the home, there is a need to better understand food waste in this setting [10].

Hospitality professionals and academics who have researched food waste in the hospitality sector recognise the significant financial, environmental and positive public perception opportunities that can be gained from food waste mitigation [11,12]. In fact, for every dollar invested in activity to reduce food waste, the hospitality sector can realise 14 dollars of benefit [13]. In the United Kingdom (UK), the Waste and Resources Action Programme (WRAP) estimates that food wasted by the hospitality and food service sector amounts to approximately £3.18 billion per annum [14]. About 75% of food waste in the food service sector is avoidable; therefore, there is a significant opportunity to reduce food waste at this stage of the food supply chain [15]. To unlock this opportunity, it is crucial that we understand both the drivers of food waste in the sector and the underling motivations that can be leveraged to elicit positive behaviour change [16].

In France, hospitality and food service waste amounts to approximately 15% of total food waste [17], and in Italy, 21% of total food waste occurs in restaurants alone [5]. In the UK, 18% of food purchased by hospitality and food service businesses is wasted, compared to 16% in households [14]. In 2020, Parry et al. modelled hospitality and food service food waste in the UK for 2018 [14] using the most recent quantitative data for the sector collected by WRAP between 2009 and 2011 [18]. Modelling indicated that a total of 1,098,000 tonnes (i.e., 12% of total food waste in the UK) per annum are wasted by the sector, 600,000 tonnes of which are wasted in restaurants (including pubs and hotel restaurants) [14]. Given the significant quantities of food wasted in this sector, WRAP launched a three-year voluntary agreement in 2012, known as the Hospitality and Food Service Agreement, to encourage hospitality businesses to reduce food waste [19]. To provide support to the sector, the Hospitality and Food Service Action Plan was launched in April 2019, outlining specific food waste reduction actions out to 2026 [20]. WRAP also initiated the "Guardians of Grub" campaign in May 2019, a partnership with the sector to support businesses to measure and reduce food waste in their operations [21]. These efforts are focused on how food service operators can reduce the food waste they produce. Importantly, however, not all food waste is produced in the kitchen: a significant amount is produced by consumers.

A major cause of food waste in restaurants is edible portions of food served to consumers that are left uneaten, known as "plate waste" [11,15,22]. WRAP found that 45% of food waste in the food service sector happens during preparation, 21% is from spoilage and 34% is plate waste [23]. Plate waste, in particular, is almost entirely avoidable, as it is the food served to customers intended for consumption [11,24–26]. For example, Silvennoinen et al. (2015) noted the greatest proportion of waste arising in Finnish restaurants is plate waste (as opposed to kitchen waste or serving waste), and Grunders (2012) [22] and Betz et al. (2015) [11] found the same in the United States due to large portion sizes [27].

In New Zealand, the setting for the present study, research in 2018 found that cafés and restaurants create 24,366 tonnes of food waste each year, of which 61% is avoidable [28]. A survey commissioned by the Restaurant Association found that 45% of New Zealanders consume food in out-of-home settings 1–3 times per week [29]. Due to the frequency with which New Zealanders are consuming food in cafés and restaurants, the contribution of out-of-home plate waste to total food waste in New Zealand is likely to be significant [9].

Filimonau et al. (2020) highlight consumer behaviour as a key target area for food waste reduction in restaurants [15]. However, in order to influence consumer behaviour and design effective interventions, we must understand situational factors and underlying attitudes that cause consumers to waste food out of the home [30]. There is limited literature on the relationship between consumer attitudes and sustainable behaviour in restaurants [31]. As consumer behaviour is a key driver of food waste in restaurants [32], it makes sense to engage consumers in the solutions for mitigating food waste [33]. In order to do this effectively, it is important to understand consumer awareness of

issues associated with food waste as well as their underlying attitudes in order to effectively motivate behaviour change [34].

Chen and Jai (2018) indicated that the large amount of customer plate waste in restaurants may be in part attributable to a lack of customer awareness of food waste issues [35,36]. Graham-Rowe et al. (2014) indicate that although consumers are aware of the cost of wasting food, the environmental and social consequences of food waste are often less well understood [37]. Laven (2017) recommends that research efforts focus on the relationship between environmental awareness and food waste behaviour [30]. The way food waste reduction messages are communicated is important to encourage proenvironmental behaviour change [27]. While messaging campaigns at a food service level can increase consumer awareness and encourage behaviours to reduce food waste [27], few studies have yet attempted to understand consumer attitudes concerning food waste when eating out of the home [5].

Thus, in order to inform effective targets for consumer food waste reduction messaging in an out-of-home setting, a nationwide questionnaire looking at out-of-home food waste in New Zealand was conducted. This study hypothesised that out-of-home food waste may be influenced by characteristics of the eating occasion, as indicated by Lorenz-Walther et al. (2019) [25] and WRAP (2013) [18]. Lorenz et al. (2017) [38] found the situational factors of taste perception and portion size to be correlated with plate waste and recommend investigation of the effect of additional situational factors. It has also been hypothesised that that there may be specific motivational factors that encourage the reduction of food waste [25,34]. The present study sought to understand the effect of situational dining and motivational factors on out-of-home plate waste in cafés and restaurants through analysis of the nationwide survey, as these factors are important precursors to designing targeted interventions [16,25,30,34]. Findings from this study will support restaurant and café managers in communicating with consumers on issues they are engaged with to lead to more effective behaviour change in terms of plate waste reduction.

## 2. Methods and Materials

This study focuses on three sections of the nationwide "Consumer Food Waste in Restaurants/Cafés" questionnaire designed to understand consumers' practices and attitudes towards plate waste in cafés and restaurants in New Zealand. The questionnaire was conducted in accordance with the University of Otago's code of research ethics (reference number: 14/06B). The original questionnaire was comprised of 70 questions on self-reported out-of-home plate waste behaviour, intentions not to waste food, subjective norms around leaving food on the plate, moral norms about wasting food, perceived behavioural control (PBC) in finishing the food on the plate when eating out, planning routines before eating out and ordering habits. All questions were adapted from previous surveys, the majority from WRAP [39].

Participants were required to be over the age of 18 and have eaten out in a restaurant or café in the past month. Respondents who were employed in a restaurant or café were excluded from participating in the survey. The sample aimed to be nationally representative of the New Zealand population over the age of 18 in terms of gender, age and income. The questionnaire was administered to participants using the online database of Research Now market research company. Surveys were emailed to Research Now members; 1378 potential respondents were approached, of which 1059 respondents agreed to participate in the survey. After taking into account missing values and unengaged responses, a total of 1004 respondents completed the entire survey, implying a response rate of approximately 72.6%. Less than 1% of participants identified as the gender "other". In order to ensure anonymity, these participants were excluded from analysis due to the very small sample size, and the final sample was 1001 participants. Participants were able to withdraw from the study at any time and without any disadvantage.

Data collection was held over a two-week period using Qualtrics survey software and exported to Microsoft Excel. The subset of the questionnaire of interest was extracted from the full dataset by the first author of this study and analysed using Stata. The selected sections included demographic information, dining factors and their relationship with food waste and motivational factors that may

encourage consumers to reduce their out-of-home food waste. Questions are outlined in Table 1. Participants were asked to think about their most recent dining experience at either a restaurant or café (excluding takeaways and fast food chains such as McDonald's). They were asked to write the name of the restaurant or café, and the survey software then populated the remaining questions with this name to help participants stay focused on this specific dining experience.

**Table 1.** Questions extrapolated from the wider national survey "Understanding consumer's restaurant/café plate waste" analysed in this study.

| |
|---|
| **Q. What was the main purpose of this visit?** |
| Functional (needed food); Social (special occasion, work, catch up with friends/family); Other—please specify. |
| **Q. In your opinion, was this restaurant or café?** |
| Cheap; Midrange; Expensive. |
| **Q. What type of meal was consumed at the restaurant or café?** |
| Breakfast; Lunch; Dinner; Snacks (e.g., morning/afternoon tea); Other—please specify. |
| **Q. What length of time was spent at the restaurant or café?** |
| Up to 30 min; 30 min to just under an hour; 1 h to 1 h 30; More than 1 h 30. |
| **Q. Please indicate how effective the following motivations would be in convincing you to reduce your restaurant or café food waste:** |
| *(1) Save money (wasting food wastes money);* <br> Very ineffective; Ineffective; Somewhat ineffective; Neither effective nor ineffective; Somewhat effective; Effective; Very effective. <br> *(2) Save the planet (wasting food wastes natural resources);* <br> Very ineffective; Ineffective; Somewhat ineffective; Neither effective nor ineffective; Somewhat effective; Effective; Very effective. <br> *(3) Save hungry people (some food wasted could feed those in need);* <br> Very ineffective; Ineffective; Somewhat ineffective; Neither effective nor ineffective; Somewhat effective; Effective; Very effective. <br> *(4) Save guilt (some people regret and are frustrated when they waste food).* <br> Very ineffective; Ineffective; Somewhat ineffective; Neither effective nor ineffective; Somewhat effective; Effective; Very effective. |

The four motivators for food waste reduction of "saving money", "saving guilt", "saving the planet" and "saving hungry people" have been investigated in various contexts in the literature, including at a household level by WRAP in the UK [39], in the US [35] and in New Zealand [40] and also from a café and restaurant management perspective in New Zealand [41]. It is useful to investigate these motivators across the food supply chain and from various perspectives to be able to draw comparisons.

The demographic characteristics of age, education and income are reported as the number (n) and percentage (%) of the sample population by males and females, respectively.

A test of proportions was used to calculate the proportion of participants who reported leaving food on their plate in each dining situation and 95 percent confidence intervals (95% CI) for the proportions in each category. A chi-squared test was used to determine the *p* value for the difference between proportions of consumers who left food on their plate in each subcategory for the four situational dining factors of interest. A *p* value of $p < 0.05$ was deemed as statistically significant. The proportion of participants who reported that the given motivational factors (saving money, saving the planet, saving hungry people, preventing guilt) may encourage them to reduce their food waste was analysed by sex and by age. The full survey used a five-point Likert scale for individuals to nominate the likelihood of each motivational factor encouraging them to reduce out-of-home food waste. The scale ranged from "ineffective" through to "very effective". In this study, the categories of

"somewhat effective", "effective" and "very effective" were collapsed into a single indicator to identify any reported degree of effectiveness associated with a given motivator.

## 3. Results

Table 2 presents demographic characteristics of participants by sex. More females participated in the survey than males; however, for each sex, the percentage of participants appeared to be relatively similar for each category, with the exception of the 25–34 age group, where the percentage of women (22.7%) was greater than that of men (16%), and >55, where the percentage of men (32.8%) was greater than that of women (22.5%) in this category. The percentages across subcategories were similar between males and females for level of education. For income level, a higher percentage of women (38.5%) earned less than $40,000 per annum compared to men (21.8%), and a greater percentage of men (27.3%) earned $80,000 or more per annum than women did (14.3%).

**Table 2.** Demographic characteristics of participants (n = 1001).

| Characteristic | n [1] (%) | |
|---|---|---|
| | Males (n = 476) | Females (n = 525) |
| Age (mean ± SD) | 44.9 ± 13.0 | 41.8 ± 13.1 |
| 18–24 years | 42 (8.8) | 64 (12.2) |
| 25–34 years | 76 (16.0) | 119 (22.7) |
| 35–44 years | 102 (21.4) | 105 (20.0) |
| 45–54 years | 100 (21.0) | 119 (22.7) |
| >55 years | 156 (32.8) | 118 (22.5) |
| Level of education | | |
| Schooling incomplete | 48 (10.1) | 39 (7.4) |
| Completed secondary | 98 (20.6) | 113 (21.5) |
| Tertiary certificate | 159 (33.4) | 160 (30.5) |
| University degree | 171 (35.9) | 213(40.6) |
| Income level (NZ$) | | |
| <40,000 | 104 (21.8) | 202 (38.5) |
| 40,000–79,999 | 174 (36.6) | 165 (31.4) |
| >80,000 | 130 (27.3) | 75 (14.3) |
| Prefer not to say | 68 (14.3) | 83 (15.8) |

[1] n = 3 identified as the gender "other". These participants were excluded from analysis in the present study to protect anonymity.

The proportions of participants who left food on their plate under the situational dining factors of "meal purpose", "price", "occasion" and "length" are highlighted in Table 3. The proportion of people who left food was not significantly different between participants who reported eating for a functional purpose compared with those who ate for a social purpose or other (*p* = 0.237). Meal price appeared to have a statistically significant effect on the proportion of people who left food on their plate (*p* = 0.009). A greater proportion of people who consumed expensive meals (46.7%) left food on their plates compared with those who dined in cheaper or midrange restaurants. There also appeared to be a statically significant difference (*p* < 0.001) across eating occasions, with a greater proportion of people dining out for dinner or in the "other" category reporting to have left food on their plates. The relationship between length of the meal and the occurrence of plate waste also appeared to be significantly different (*p* < 0.001) depending on meal length, with the greatest proportion of participants reporting food waste when the meal duration was greater than one and a half hours.

**Table 3.** Factors affecting the proportion of participants who left food on their plate during the last occasion when participants dined out of the home.

| Factors | Proportion (95% CI) | *p* Value [1] |
|---|---|---|
| **Purpose of eating** | | |
| Functional | 0.308 (0.256–0.366) | |
| Social | 0.361 (0.326–0.398) | 0.237 |
| Other | 0.303 (0.202–0.427) | |
| **Price of meal** | | |
| Cheap | 0.280 (0.219–0.352) | |
| Midrange | 0.342 (0.308–0.377) | 0.009 |
| Expensive | 0.467 (0.366–0.571) | |
| **Eating occasion** | | |
| Breakfast | 0.275 (0.232–0.324) | |
| Lunch | 0.244 (0.160–0.353) | |
| Dinner | 0.427 (0.384–0.471) | <0.001 |
| Snack | 0.175 (0.098–0.292) | |
| Other | 0.444 (0.134–0.805) | |
| **Length of meal (h)** | | |
| <0.5 | 0.228 (0.170–0.300) | |
| 0.5–1.0 | 0.286 (0.239–0.338) | <0.001 |
| 1.0–1.5 | 0.391 (0.341–0.444) | |
| >1.5 | 0.452 (0.383–0.525) | |

[1] Determined using chi-squared test.

Figures 1 and 2 report the proportion of participants who identified given motivational factors to be effective in encouraging them to reduce out-of-home food waste in cafés and restaurants. Figure 1 presents this data by sex. Overall, a greater proportion of females believed that the given motivational factors would encourage them to reduce out-of-home food waste. Across all motivational factors, saving money was reported by the greatest proportion of both males (0.708) and females (0.832) as a factor that would encourage them to reduce their out-of-home food waste. For each sex, similar proportions were motivated to reduce plate waste by saving the planet and saving hungry people (0.680 and 0.709 of females, respectively, and 0.544 and 0.592 of males, respectively).

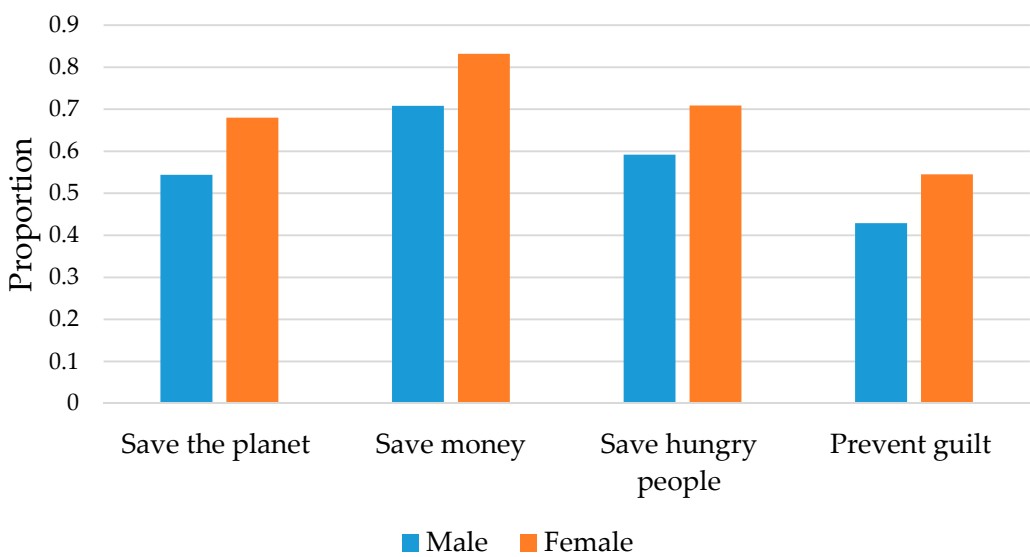

**Figure 1.** Factors that would encourage participants to reduce out-of-home food waste, by sex.

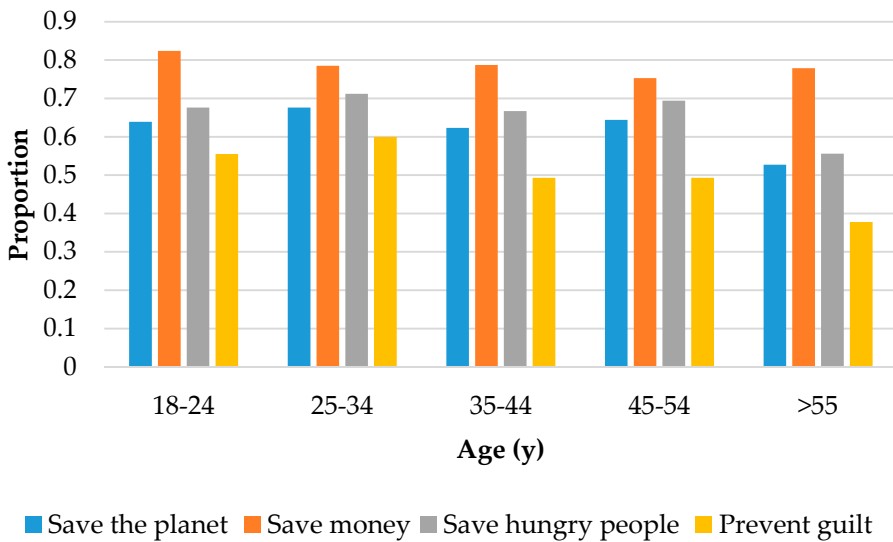

**Figure 2.** Factors that would encourage participants to reduce out-of-home food waste, by age.

Figure 2 presents the given motivational factors stratified by age. Across all age groups, the greatest proportion of people was motivated to reduce out-of-home food waste by saving money (0.824, 0.785, 0.787, 0.753 and 0.779, respectively, across age groups). A similar proportion of people who thought that the given motivational factors would encourage them to reduce their out-of-home food waste was reported across all age categories, with the exception of the >55 category. Only a small proportion of people in the >55 age group identified the given factors as effective in motivating plate waste reduction, with the exception of saving money as a motivating factor.

## 4. Discussion

The greatest proportion of people who left food on their plate was those who were consuming an expensive meal (Table 3). These findings were echoed by Beretta and Hellweg (2019), who found that the largest amounts of food waste were measured in luxury restaurants across Germany, Austria, Switzerland, Finland and the UK [42]. People dining in expensive establishments may have more disposable income, and the value they place on food for survival may be less than, for example, someone facing food insecurity would. Those who do not need to worry about availability of food in the future are less likely to value it and, therefore, more likely to waste food [43]. This may explain why the greatest proportion of "food wasters" reported that they were consuming an expensive meal. Customers at high-end restaurants may also be more concerned about overconsumption; this health-related concern may have contributed to greater plate waste [44]. Offering consumers a choice in portion size, the ability to order meals course by course or modifiable sides as an optional extra could be well-suited to enabling plate waste reduction in more expensive restaurants.

The proportion of people who left food on their plate was the greatest if the meal lasted more than 1.5 h (Table 3). During a longer meal, more conversation may take place, and the eating pace may be slower. When consumers take longer over their meal, satiety can set in before the meal is finished, resulting in a higher likelihood of plate waste [45]. Contrary to this, studies in a cafeteria food service setting indicated that time pressure (i.e., a shorter meal duration) may result in increased plate waste [46,47]. While a review of these past studies would seem to suggest, then, that the optimal meal duration to minimise food waste might lie in between a short meal and a long meal, the picture seems to be more complex. Lorenz-Walther et al. (2019) [25] and Lorenz et al. (2017) [38] found that time pressure was not correlated with plate waste. Lorenz et al. (2017) suggest that under time pressure, the influence of motivational behavioural factors diminishes, and situational dining factors have an effect on plate waste, whereas when time pressure is not present, attitudes have more

influence [38]. Also contradicting the idea that a medium meal duration is optimal, our findings showed that the proportion of people who left food on their plate was the greatest if the meal lasted more than 1.5 h. This has an important management implication, as it indicates that interventions targeting consumers' underlying motivations to reduce plate waste may be more effective when meal duration is longer, which, as this study found, is correlated with plate waste. Therefore, interventions targeting consumer motivations in this setting may have substantial impacts on plate waste reduction.

A greater proportion of people left food on their plates when dining out for dinner (Table 3). As most out-of-home food consumption occurs at dinner (48%) [29], food waste reduction efforts should be targeted at establishments serving evening meals. To the best of our knowledge, the relationship between the proportion of "food wasters" and mealtime has not been previously researched. We hypothesise that consumers may leave more food on their plates when dining out for dinner, as serving sizes tend to be larger. We recommend future research test the effect of availability of reduced portion sizes (i.e., half-portions) on plate waste. Future studies may also investigate whether consumers dining out for dinner are more likely to dine in expensive establishments, spend longer over their meal and be eating for a social purpose. A limitation of this analysis is that is does not look at whether these situational dining factors have a multifactorial effect or are working in isolation. However, it does appear that situational factors have a significant impact on plate waste, as has also been noted by Lorenz-Walther et al. (2019), who support the assumption that personal attitudes and situational dining factors contribute to consumers' plate waste [25].

The four motivational factors for reducing food waste assessed in this study were saving money, saving the planet, saving hungry people and preventing guilt (Table 1). These factors have also been studied in the UK in the context of household food waste [39], in a New Zealand household context [40], from restaurant staff members' perspectives [41] and more generally in a US consumer food waste survey [35]. In all cases, saving money was the strongest motivator for reducing food waste, which was also reflected in the results of this study. The motivator of saving money is used widely in food waste reduction initiatives [9], as it is a motivator that traverses different stages of the food supply chain and resonates with various actors along it.

Overall, a greater proportion of women reported that the given motivational factors would encourage them to reduce their plate waste (Figure 1). Further analysis could look into social cognitive theory in the context of food waste to understand if women in the study were at a more advanced stage of change than men [48]. Perhaps women were thinking more about reducing their food waste, and this contributed to them identifying that more of the motivational factors could have a positive effect on their behaviour. When stratified by age (Figure 2), saving money was also the most motivating factor that people identified for reducing their out-of-home food waste. Promotions that offer cost savings—for example, offering a smaller portion at a cheaper rate—are likely to resonate with a large proportion of customers. This strategy was suggested by WRAP as a technique to reduce food waste resulting from large portion sizes. Pairing the reduction in portion size with the motivational factor of saving money could appeal to consumers' values [18]. Aschemann-Witzel (2015) commented that sociodemographic variables do not seem to play a role in explaining food waste [49]. This concurs with the findings of this study, where motivators were prioritised in the same order irrespective of age and sex. Therefore, we conclude that it is not necessary to tailor food waste reduction interventions aimed at reducing plate waste in the hospitality sector to specific age groups or by sex.

WRAP's research into motivators to reduce household food waste also showed saving money as the strongest motivator for consumer food waste reduction. WRAP's regular survey on food waste found that 41% of the population was motivated "a great deal" by saving money, and a further 34% were motivated "a fair amount" [39]. In 2009, WRAP also surveyed 1153 restaurant customers about out-of-home food waste in the UK. When asked about reasons for concern about food waste, 72% of respondents believed it was a waste of money, 22% commented that it made them feel guilty and 21% identified that it was bad for the environment [50]. WRAP developed the "Resource pack for Hospitality and Food service sector" to support businesses in engaging with consumers on reducing their plate

waste [51]. This resource pack leverages the motivational factor of saving money to encourage the desired outcome of reducing customer plate waste. The resource provides guidance on measuring plate waste because "what gets measured gets managed" [3]. It also provides guidance on the best way to frame messages to customers and emphasises the importance of positive messaging and providing customers with simple solutions. We recommend the adoption of guidance included in this resource pack by cafés and restaurants globally.

In a representative nationwide survey about household food waste with respondents from 1300 New Zealand households, 84% of respondents reported being motivated to reduce food waste by the possibility of saving money. The motivators of saving guilt (69%), the environment (60%) and hungry people (49%) were also identified, but to a lesser extent [40]. As a result, New Zealand's Love Food Hate Waste campaign focuses on framing food waste reduction as a cost-saving activity in order to connect with this known motivator for behaviour change [9].

Research conducted into restaurant and café food waste in New Zealand in 2017 asked an appropriate staff member from 31 participating restaurants and cafés to what degree the same four motivational factors would motivate them to reduce food waste at work. Saving money was also identified as the strongest motivational factor, with 97% of participants rating this as "very" or "extremely" important. Many café and restaurant staff identified saving the planet and saving hungry people as important, similar to consumer findings in this study; however, café and restaurant staff ranked saving the planet as slightly more motivational compared with consumers, who were slightly more motivated by saving hungry people. Saving guilt was also the lowest-ranked motivator by restaurant and café staff [41], which is in agreement with consumer opinions elicited in this research.

These are interesting findings, as they illustrate that both café and restaurant staff and customers are motivated by the factor of saving money in terms of food waste reduction. Both are also motivated by social good factors, such as saving the planet and saving hungry people, but less motivated in terms of preventing guilt. These findings indicate that behaviour change campaigns should focus on the benefits that can arise from food waste reduction rather than highlighting the negative impacts of food waste.

Similarly, Neff et al. (2015) surveyed 1010 consumers in the US on the importance of motivational factors on reducing food waste using a four-point Likert scale from "not important at all" to "very important". As found in the present study, the most important motivation was saving money [35]. Interestingly, 22% of those surveyed ranked environmental motivations as "not at all important", and only 10% ranked them as "very important". The lower priority respondents placed on environmental motivators has also been found in other studies but to a slightly lesser extent (i.e., 20% of participants were highly motivated by environmental concerns in the UK and US) [35]. A survey in Canada showed that 83% of participants believed food waste is a social problem, 72% identified it as an economic problem and 68% as an environmental problem [52].

The present study also found that environmental motivations ranked third behind economic and social motivators. Often, organisations who are communicating messages around food waste reduction are highly motivated by the environmental outcomes of food waste reduction and lean towards communicating messages through this lens. However, perhaps a shift in communication from what motivates the organisation to what motivates the consumer is needed to allow these messages to be delivered more effectively [35]. WRAP notes that although food waste reduction leads to beneficial environmental outcomes, consumers may not perceive actions to reduce food waste as "environmental" or "sustainable". For this reason, WRAP has framed the Love Food Hate Waste campaign to focus on encouraging food use instead of waste reduction [49], and the hospitality and food service resource mentioned previously focuses on the motivation of saving money [51].

WRAP comments that "the fact that environmental concerns and those associated with food shortages elsewhere in the world have less weight placed on them indicates that the link between food waste and environmental impact is not firmly established in people's minds, even though the impact on the environment and the world's resources is considerable" [53] (p. 11). Neff et al. (2015) hypothesised

that a lack of knowledge about the association between wasting food and environmental harm may contribute to "saving the planet" being identified by fewer consumers as a motivation to reduce food waste [35]. Consumers can draw clear associations with food costing money and lack of food causing hunger, but the relationship with the environment is more difficult for consumers to personally relate to. WRAP suggests that engaging solely on proenvironmental or prosocial values as a single intervention is unlikely to bring about the desired food waste reduction behaviour change unless consumers first understand the links between food consumption and the environment [39]. Understanding consumer attitudes is complex, and multiple values, social factors and habits come together to bring about action [39]. Aschemann-Witzel et al. (2015) recommend that although saving money may be a key driver of food waste mitigation behaviours, initiatives that tap into consumers' underlying values and beliefs (i.e., ethical, environmental, religious beliefs) are likely to strengthen attitudes about avoiding food waste [49]. Secondi et al. (2019) investigated consumer attitudes around perceptions of reducing food waste as an economic opportunity, the correct behaviour toward food security or whether it had a negative impact on the environment. Results showed that those who recognised the economic, social and environmental associations with food waste were significantly less likely to waste food out of the home [5].

There is a clear need for education about the environmental impacts of wasted food [35]. Filimonau et al. (2019) note that without consumer awareness of both environmental and social impacts of food waste, it is difficult to promote food waste reduction. Conveying information about these impacts is important in raising consumers' awareness and motivation to reduce food waste [34]. Laven (2017) found that awareness of the impact of food waste on the environment supported positive behaviour change and hypothesised that restaurant managers are likely to see changes in food waste behaviour if they raise customer awareness of these impacts [30].

When it comes to effectively communicating information about the connection between food waste and the environment, evidence shows that positively framed messages (i.e., informing on the environmental benefits resulting from reducing food waste) are likely to resonate with consumers more than negatively framed messages (i.e., the pollution caused by food waste releasing methane) [54]. Chen and Jai (2018) surveyed 169 participants to investigate the effect of positive and negative environmental messaging, and results indicated that consumers responded favourably to positively framed messages and, as a result, were more likely to make efforts to reduce food waste when dining in restaurants [27].

Lorenz-Walther et al. (2019) tested various assumptions on the impact of information provision about a food service provider's efforts to minimise food waste on consumer plate waste [25]. The information was presented in the form of a poster and stated the restaurant's commitments to reducing food waste, encouraging the customer to help by only taking as much as they needed, asking for smaller portions and choosing alternatives if their first choice was not available. Interestingly, customers did remember that they had read the information provided and what it stated; however, this was not reflected in a statistically significant impact on behaviour change overall. The authors found that in some cases, information provision resulted in stated and observed reductions in plate waste, and in other cases (i.e., being asked to choose an alternate dish), it resulted in additional plate waste [25]. One potential reason that the overall information provision did not result in significant behaviour change could be that the information was not tapping into multiple value-based motivations for change; it focused solely on behaviours and not values. It could be useful to test whether information provision on the cost savings of reducing food waste (i.e., a smaller portion is cheaper), the environmental benefits of reducing food waste or the social benefits of providing food to those in need might impact plate waste. Lorenz-Walther et al. (2019) commented that there was evidence to support that campaigns that communicate environmental messages can lead to a reduction in plate waste in out-of-home settings [25].

Ideally, this positive messaging should be demonstrated to the consumer in the practices of the café or restaurant. One way of doing this is through embedding corporate social responsibility (CSR)

throughout operations. Dief and Front (2010) found that consumers were more likely to emulate behaviours they saw demonstrated in the establishment [55]. Filimonau et al. (2020) also suggest food waste reduction should be adopted as a corporate target by food service establishments and monitored by senior management [15]. Chen and Jai (2018) suggest that if restaurants and cafés lead by example, consumers will be more likely to act upon their requests to reduce food waste [27]. Aschemann-Witzel et al. take things one step further and recommend that the focus should be shifted towards eating the food instead of discussing wastage [49]. Love Food Hate Waste ran the "ComplEAT" campaign for households, encouraging consumers to eat all edible parts of food. Perhaps restaurants could frame messages in a similar light, presenting consumers with solutions for how to "ComplEAT" when dining out of the home by asking for a doggy bag, asking for a smaller portion, ordering an entrée instead of a main dish or placing orders course by course instead of ordering all courses at the start of the meal. The practical solutions mentioned above are just a few examples of how restaurants can support consumers to reduce plate waste when eating out of the home.

In the face of the COVID-19 pandemic and its impact on the hospitality sector, it will become increasingly pertinent for cafés and restaurants to employ cost-saving measures; addressing food waste is a practical action to save businesses money [56]. The economic impact of COVID-19 is wide-reaching, and saving money is also important for consumers. Food redistribution applications are a solution that can support both the hospitality sector and people [5]. Issues such as food insecurity have been exacerbated, and wasting food is less acceptable than ever. As the hospitality sector redefines itself, tapping into consumer values will be crucial. Reducing food waste is a tangible way the hospitality sector can connect with and act on these consumer values. Consumers are now in a state of flux, and they are more receptive to change. This is an opportunity for cafés and restaurants to make zero food waste the "new normal", offer different portion sizes, use apps to offer discounted food to customers before they close and connect with food rescue organisations to distribute food to those in need.

## 5. Conclusions and Limitations

Situational dining factors appear to influence out-of-home food waste in cafés and restaurants. A greater proportion of New Zealand consumers reported leaving food on their plate when dining out if the meal was more expensive, lasted longer and when dining out occurred at dinnertime. A limitation of this analysis is that is does not look at whether these situational dining factors have a multifactorial effect or are working in isolation, and thus, research that further explores the influence of situational dining factors on food waste is warranted. For example, to the best of our knowledge, the relationship between the proportion of "food wasters" and mealtime has not been previously researched. We hypothesise that consumers may leave more food on their plates when dining out for dinner, as serving sizes tend to be larger. We recommend future research test the effect of the availability of reduced portion sizes (i.e., half-portions) on plate waste. Future studies may also investigate whether consumers dining out for dinner are more likely to dine in expensive establishments, spend longer over their meal and be eating for a social purpose. The present analysis did not link an individual's demographics or responses for situational factors with the motivators that same individual identified as most important; future studies may wish to investigate the relationship between these variables. Another limitation of this study is that while we have identified several important trends—e.g., that a greater proportion of women reported that the given motivational factors would encourage them to reduce their plate waste (Figure 1)—the purely quantitative nature of our survey did not allow for exploration of why this was the case. Thus, further analysis could investigate qualitatively, for example, if women in the study were thinking more about reducing their food waste and this contributed to them identifying that more of the motivational factors could have a positive effect on their behaviour. Employing social cognitive theory to help understand stages of change could be a fruitful avenue for exploration here [48].

In terms of implications, irrespective of sociodemographic variables, New Zealand consumers are highly motivated to reduce out-of-home food waste to save money. Previous studies found

New Zealand café and restaurant operators are also motivated by cost savings [41]. Therefore, in the first instance, initiatives aimed at reducing food waste should be framed as a cost-saving opportunity, as this motivation resonates with both the business communicating the message and the consumer. We recommend the broad adoption of the WRAP hospitality and food service resource pack [19]. As acknowledged in the literature, interventions that tap into multiple values, including environmental and social values, are even more likely to elicit behaviour change [5,39,49]. Therefore, it is recommended that future food waste reduction initiatives tap into consumers' financial, social and environmental values. Awareness of food waste as an issue is an important precursor to motivating out-of-home food waste reduction [5,27,30,34,35]. Positively positioned messages are more likely to generate positive consumer responses and translate into behaviours to reduce food waste [27]. If restaurants and cafés can demonstrate their commitment to food waste minimisation in a way that appeals to consumers' values—for example, they save money, or they see food being donated to feed hungry people—this will further build awareness of and motivation for food waste reduction in a positive way.

**Author Contributions:** Conceptualization, F.G.-S. and M.M.; Data curation, F.G.-S.; Formal analysis, F.G.-S.; Funding acquisition, M.M.; Investigation, F.G.-S.; Methodology, F.G.-S., R.M. and M.M.; Project administration, M.M.; Resources, M.M.; Software, R.M.; Supervision, M.M.; Validation, R.M.; Visualization, F.G.-S., R.M. and M.M.; Writing—original draft, F.G.-S.; Writing—review & editing, F.G.-S., R.M. and M.M. All authors have read and agreed to the published version of the manuscript.

**Funding:** This research received no external funding.

**Conflicts of Interest:** The authors declare no conflict of interest.

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
