# Peer review of "Understanding the Effect of Dining and Motivational Factors on Out-Of-Home Consumer Food Waste"

_sustainability, doi:10.3390/su12166507_

Round 1

Reviewer 1 Report

This is an interesting study in the domain of food waste behavior out-of-home.

I have made several suggestions that the Authors may want to consider in revising their paper.

METHODS AND MATERIALS

This section should be supplemented with information regarding the response rate.

The design and procedure should be provided as well.

The Authors have not specified if the present study has been approved by a research ethics committee.

RESULTS

Because 3 participants (identified as the gender ‘other) were excluded from analysis the total number of the participants is 1001. Therefore, in abstract and in the main manuscript text the number of participants should be corrected.

I propose to add information about mean age (± SD) of females and males, separately.

DISCUSSION

The Authors should include the limitation of the study.

Author Response

Thank you for your thorough review. Please see the attachment for our considered responses.

Reviewer 2 Report

The study addresses an important issue that the authors point out has been neglected in prior study. In addition to the results that are presented, this contribution, as the authors acknowledge, therefore serves as a starting point, and guide, for future research.

In general, the paper was quite nicely written, but there are a couple of minor issues in the specific comments that probably need some attention. In the following, I've also pointed out some typos but by no means all, so it certainly needs a proofread.

Specific comments:

Abstract: I think an abstract should always start with a problem statement followed by the aims. It's also not yet obvious what is meant by 'plate waste' In other words, I think the first 2 (or maybe 3) sentences of the abstract are missing.

Lines 7-9: This is counter-intuitive, which makes it even more important to tell the reader what plate waste is.

Line 29: countries

Line 36: "After food wasted in the home, food wasted out-of-home...". Are there more than these two options? 

Line 46: amounts to

Line 179: delete 'of'.

Line 213: so, should a recommendation be that only higher-end restaurants should serve smaller portions?

Line 217: Do fruit peelings and bones count as plate waste? I thought plate waste was leaving edible food on the plate.

Line 230: To me, this suggests that time pressure (from not having enough time) makes people forget other motivations, but that if people have too much time, they are no longer hungry and don't finish their meals. So the optimum must lie somewhere in between. 

Line 272: I think there's a point that might have been missed. People who eat in expensive restaurants for dinner (remembering that 'special occasions' was another category) are probably wealthier than those who choose budget restaurants, so saving money is probably a less important factor for that demographic.

Lines 287-288: I think this is the result that the previous paragraph is discussing? If so, it should probably be at the beginning of it rather than coming afterward.

Line 346: the year is missing from the citation.

Lines 352-252: needs commas.

Lines 359-363: needs commas.

Line 375: Is 'perverse' the right word?

Line 396: ....mail, or placing....

Lines 401-405: sure, but what has this got to do with the four motivations for reducing plate waste? I suggest deleting these lines and keeping them for another paper.

Lines 412-415: sorry, but I'm not convinced with this argument. People who buy restaurant food at midnight are not necessarily the same people who eat dinner in expensive restaurants (the main plate wasters). I think initiatives to feed the hungry (the 2nd strongest motivation) would carry more weight.

Lines 418-419: This sentence with the citation from Hanson and Mitchell comes as a bit of a surprise so probably needs to be in the introduction as well. It can be given a little more explanation in the intro section. Right here, it opens more questions than it answers.

Lines 450-453: "....or they see food being donated and delivering social good,....". I think you need to be more specific. In the question, it was about feeding hungry people. Of course, that delivers social good, but the research question was more specific.

Author Response

(The authors gave the same response as above.)

Round 2

Reviewer 1 Report

In my opinion, the manuscript has been largely improved and it is suitable for publication in "Sustainability".

Author Response

Thank you for your careful consideration and feedback on this paper. Please find the updated manuscript attached.